# Role of Diacylglycerol Kinases in Acute Myeloid Leukemia

**DOI:** 10.3390/biomedicines11071877

**Published:** 2023-07-01

**Authors:** Teresa Gravina, Chiara Maria Teresa Boggio, Elisa Gorla, Luisa Racca, Silvia Polidoro, Sara Centonze, Daniela Ferrante, Monia Lunghi, Andrea Graziani, Davide Corà, Gianluca Baldanzi

**Affiliations:** 1Department of Translational Medicine, University of Piemonte Orientale, 28100 Novara, Italy; teresa.gravina@uniupo.it (T.G.); chiara.boggio@uniupo.it (C.M.T.B.); luisa.racca@uniupo.it (L.R.); silvia.polidoro@uniupo.it (S.P.); daniela.ferrante@uniupo.it (D.F.); andrea.graziani@unito.it (A.G.); davide.cora@uniupo.it (D.C.); 2Center for Translational Research on Allergic and Autoimmune Diseases (CAAD), University of Piemonte Orientale, 28100 Novara, Italy; sara.centonze@uniupo.it; 3Department of Health Sciences, University of Piemonte Orientale, 28100 Novara, Italy; 4Division of Hematology, Department of Translational Medicine, University of Piemonte Orientale, 28110 Novara, Italy; monia.lunghi@uniupo.it; 5Department of Molecular Biotechnology and Health Sciences, Molecular Biotechnology Center (MBC), University of Turin, 10124 Turin, Italy

**Keywords:** lipid, signaling, bone marrow, monocyte, granulocyte, neutrophil, megakaryocyte, erythrocyte, inhibitors

## Abstract

Diacylglycerol kinases (DGKs) play dual roles in cell transformation and immunosurveillance. According to cancer expression databases, acute myeloid leukemia (AML) exhibits significant overexpression of multiple DGK isoforms, including *DGKA*, *DGKD* and *DGKG*, without a precise correlation with specific AML subtypes. In the TGCA database, high *DGKA* expression negatively correlates with survival, while high *DGKG* expression is associated with a more favorable prognosis. *DGKA* and *DGKG* also feature different patterns of co-expressed genes. Conversely, the BeatAML and TARGET databases show that high *DGKH* expression is correlated with shorter survival. To assess the suitability of DGKs as therapeutic targets, we treated HL-60 and HEL cells with DGK inhibitors and compared cell growth and survival with those of untransformed lymphocytes. We observed a specific sensitivity to R59022 and R59949, two poorly selective inhibitors, which promoted cytotoxicity and cell accumulation in the S phase in both cell lines. Conversely, the DGKA-specific inhibitors CU-3 and AMB639752 showed poor efficacy. These findings underscore the pivotal and isoform-specific involvement of DGKs in AML, offering a promising pathway for the identification of potential therapeutic targets. Notably, the *DGKA* and *DGKH* isoforms emerge as relevant players in AML pathogenesis, albeit DGKA inhibition alone seems insufficient to impair AML cell viability.

## 1. Introduction

Acute myeloid leukemia (AML) is an aggressive disease characterized by a heterogeneous genetic landscape, primarily affecting older individuals [1]. Despite advancements in targeted therapies, the emergence of chemoresistance and high relapse rates call for the identification of additional druggable targets. While most patients initially respond to chemotherapy, three-quarters of them will succumb to the disease within 5 years. Recent targeted therapies, such as those based on FLT3, IDH1, IDH2 and BCL2 inhibitors, have opened new avenues for treatment. However, the frequent development of drug resistance poses a significant challenge [2].

Myelodysplastic syndrome and AML share common features, including a high frequency of myeloblasts in the bone marrow (BM) or blood, which are arrested at specific differentiation stages, such as multipotent progenitors, common myeloid progenitors, granulocyte–monocyte progenitors, or megakaryocyte–erythroid progenitors [3]. AML is also characterized by genetic heterogeneity, with recurrent cytogenetic abnormalities observed in 20–30% of AML patients, along with several common mutations [4]. To address this heterogeneity, a recent classification has defined two main classes of AML: AML with well-defined genetic abnormalities and AML classified based on the differentiation state [5]. 

Diacylglycerol kinases (DGKs) are a family of lipid signaling regulators that decrease diacylglycerol signaling while simultaneously generating phosphatidic acid, a bioactive lipid. Ten mammalian DGK isozymes are present in the human genome, where multiple isoforms are co-expressed at the same time, suggesting non-redundant roles. DGKs are classified into five sub-families based on distinct regulatory domains, typically located in the N-terminal portion of the protein, preceding a conserved catalytic domain. This domain comprises a catalytic ATP binding site and accessory subdomains and is preceded by at least two C1 domains homologous to the diacylglycerol binding region of PKC. Class I DGKs (i.e., DGKA, DGKB and DGKG) possess two calcium-binding EF hands and a recoverin homology domain, making them calcium-regulated. Class II DGKs (i.e., DGKD, DGKH and DGKK) contain a phosphoinositide-binding PH domain and, apart from *DGKK*, a C-terminal sterile alpha motif for protein–protein interaction. The only member of class III DGKE shows an amphipathic helix that facilitates protein localization to membranes and confers strong selectivity for polyunsaturated fatty acids at position two of the substrate. Class IV DGKs (i.e., DGKZ and DGKI) feature ankyrin repeats for protein–protein interaction. Finally, DGKQ, which belongs to Class V, harbors three C1 domains and a Ras association domain of unknown function [6].

The activity of several DGK isoforms has been implicated in both cell transformation and immunosurveillance against tumors [7]. Strong evidence supports the involvement of DGKA activity in the development of glioblastoma [8,9], melanoma [10,11] and hepatocellular carcinoma [12]. Accordingly, DGKA transduces the proliferative effects of growth factors [13], cytokines [14] and oncogenes such as NPL–ALK [15]. Furthermore, depending on the cellular model, other DGK isoforms may be involved in tumorigenesis, as reviewed in [16]. Finally, DGKA and DGKZ function as negative regulators of the immune response by depleting diacylglycerol, thereby inhibiting the anticancer activity of lymphocytes and natural killer cells [17,18,19]. Consistently, DGKA targeting rescues defective T cell receptor (TCR) signaling strength in experimental models of X-linked Lymphoproliferative Disease 1 and Wiskott–Aldrich syndrome [20,21].

The potential therapeutic value of DGK inhibitors in the treatment of tumors and hematological diseases has led our group to develop new DGKA-specific inhibitors [22,23] and explore pan-cancer studies to identify potentially sensitive tumor types. Intriguingly, Li and colleagues reported high expression of *DGKZ* in HL-60 cells, a model of AML, and demonstrated that its knockdown reduces cell proliferation rate by inducing cell cycle arrest at the G_2_/M phase, accompanied by increased apoptosis. These effects were associated with decreased MAPK and surviving activity, as well as enhanced caspase levels [24].

In this study, we aimed to investigate the potential involvement of DGK family members in AML by conducting a detailed analysis of expression databases and examining the in vitro effect of DGKA inhibitors on leukemia cell lines. 

## 2. Materials and Methods

### 2.1. Cancer Tissue Database Exploration

GEPIA2 (http://gepia2.cancer-pku.cn/ accessed on 1 August 2022) was used to explore the TGCA database for DGK isoform expression. Data are shown as expression = −log2(TPM + 1), where TPM stands for transcripts per million [25]. 

To access the data from the BeatAML database [26] the vizome tool (http://www.vizome.org/ accessed on 1 August 2022) was used. Expression data, represented as RPKM = reads per kilobase of exon per million reads mapped, were analyzed by one-way ANOVA and visualized using GraphPad Prism.

For gene expression data in different tumor types and corresponding survival curves, we relied on the Ingenuity Pathway Analysis (IPA) Land Explorer tool (QIAGEN Inc., Hilden, Germany, https://www.qiagen.com/us/products/discovery-and-translational-research/next-generation-sequencing/informatics-and-data/interpretation-content-databases/ingenuity-pathway-analysis/; accessed on 1 May 2023). In particular, we employed the TCGA, BeatAML, and TARGET datasets to download and analyze survival curves for DGK genes within AML tumor types. In the TCGA dataset, we selected the samples related to the “LAML” tumor type. Concerning the BeatAML [26] dataset, we used the entries “LAML” and “AML with NPM1 mutation” for sample selection. As for the TARGET dataset, we considered the samples related to the “LAML” tumor type, representing in this case a set of pediatric patients with AML, from the Therapeutically Applicable Research to Generate Effective Treatments initiative (https://www.cancer.gov/ccg/research/genome-sequencing/target accessed on 1 May 2023), phs000465. Survival analysis was performed using a Log Rank Test, based on the *survdiff*() function of the statistical package R (www.r-project.org accessed on 1 May 2023). 

### 2.2. Coexpression

To identify genes positively and negatively correlated with DGK isoform expression, we performed a query on the BeatAML database from CBioPortal (https://www.cbioportal.org accessed on 1 November 2022, Acute Myeloid Leukemia—OHSU, Nature 2018 dataset) for DGK co-expressed genes. The following rules were used: for positively correlated genes, a Spearman’s correlation coefficient > 0.40 and a q-value < 0.05 were considered, while for negatively correlated genes, a Spearman’s correlation coefficient < −0.40 and a q-value < 0.05 were used. Functional analysis of the gene sets positively and negatively correlated to the DGK isoforms was performed using Metascape [27], downloading the “Top enriched pathways” and “PaGenBase” heatmaps.

Clinical data and mRNA expression (RNA-Seq RPKM) values were extracted from the same dataset through cBioPortal to obtain the expression levels of DGK isoforms for each AML subtype. Only AML subtypes with more than 5 samples and known nomenclature were retained for subsequent analysis. Boxplots were computed using the *boxplot*() function in R. 

### 2.3. Cell Lines

The HL-60 cell line was established from the peripheral blood of a 35-year-old woman with AML in 1976. It harbors a homozygous deletion of TP53, a homozygous p.Arg80Ter mutation of CDKN2A, and a heterozygous p.Gln61Leu mutation of NRAS (source: Cellosaurus Expasy, www.cellosaurus.org accessed on 1 January 2023).

The HEL cell line was established from the peripheral blood of a 30-year-old man with erythroleukemia in relapse in 1980—following treatment for Hodgkin lymphoma. It carries JAK2 p.Val617Phe and TP53 p.Met133Lys mutations (source: Cellosaurus Expasy, www.cellosaurus.org accessed on 1 January 2023).

The identity of both cell lines was confirmed by the Cell Line Authentication Test (Eurofins Genomics, Ebersberg, Germany), and the resulting markers were 100% consistent with those in the ATCC database. Both cell lines were cultured in RPMI 1640 medium supplemented with 10% fetal bovine serum. Peripheral blood lymphocytes (PBLs) were isolated from healthy human buffy coats obtained from anonymous donors; this service was provided by the Transfusion Service of Ospedale Maggiore della Carità, Novara, Italy. PBLs were isolated by Ficoll-Paque PLUS (GE Healthcare, Chicago, IL, USA) density gradient centrifugation, washed, and resuspended at 2 × 10^6^ cells/mL in RPMI-GlutaMAX containing 10% heat-inactivated FBS (Lonza, Basel, Switzerland), 2 mM glutamine and 100 U/mL of penicillin and streptomycin (Life Technologies, Carlsbad, CA, USA). T cells were activated with 1 μg/mL anti-CD3 (clone OKT3) and anti-CD28 (clone CD28.2) human antibodies for 72 h. Activated T cells were then washed in PBS and cultured in a complete medium supplemented with 100 IU/mL rhIL-2 (PeproTech, Cranbury, NJ, USA) at 1–2 × 10^6^ cells/mL for ≥7 days, with medium changes every 2–3 days. 

### 2.4. Chemicals and Inhibitors

R59949 (Cayman Chemical Company, Ann Arbor, MI, USA), R59022 (Sigma, St. Louis, MO, USA), CU-3 (Vitasmlab, Causeway Bay, Hong Kong) and AMB639752 (Ambinter, Orléans, France) were all dissolved in dimethyl sulfoxide (DMSO) at a concentration of 10 mM and stored at −20 °C until use. The final concentration of DMSO was always <1% and equal volumes of DMSO were added to the control samples to check for potential toxicity. Unless specified otherwise, all chemicals were from Merck (Darmstadt, Germany). The activity of all inhibitors was verified in vitro, as described below and shown in Appendix A.

### 2.5. DGKA Protein Purification

A GST-tagged *DGKA* construct [28] was overexpressed in 293T cells previously cultured in 10-centimeter petri dishes. Transfection was performed using Lipofectamine 3000 Reagent (Thermo Fisher Scientific, Waltham, MA, USA) according to the manufacturer’s instructions. At 48 h post-transfection, cells were lysed in 1 mL of lysis buffer (25 mM HEPES pH 8, 150 mM NaCl, 1% Nonidet P-40, 5 mM EDTA, 2 mM EGTA, 50 mM NaF, and 10% glycerol, supplemented with 1 mM Na_3_VO_4_, 2 mM dithiothreitol and protease inhibitors) and clarified by centrifugation at 12,500 rpm at 4 °C for 15 min. Subsequently, the lysates containing GST–DGKA protein were incubated with 150 µL of glutathione–agarose beads (GE Healthcare) with gentle agitation at 4 °C. After 4 washes in lysis buffer and 2 in phosphate-buffered saline (PBS), the pulled-down proteins were eluted with 100 µL elution buffer (100 mM Tris HCl pH 8.0, 10 mM NaCl, and 5% glycerol, supplemented with fresh 2 mM DTT and glutathione 10 mM). 

### 2.6. DGKA Activity Assay

DGK kinase assay was performed using the DGKA Kinase Enzyme System (Promega, Madison, Wisconsin; Catalogue #: VA7606) and the ADP-Glo Kinase Assay Kit (Promega; Catalogue #: V9101) according to the manufacturer’s instructions. In brief, this detection assay comprises two steps: (1) After the kinase reaction, the reagent ADP-Glo^TM^ is added to terminate the kinase reaction and deplete any remaining ATP. (2) A second reagent converting ADP to ATP is added, generating light through a luciferase/luciferin reaction. This signal is proportional to the amount of ADP present and, consequently, to the kinase activity. 

The assay was performed in a 384-well luminescent white plate in a 5 µL reaction volume. DGKA kinase reaction was carried out using DGKA, purified as described in Section 2.5, diacylglycerol as lipid substrate (diluted in the provided commercial Lipid Dilution Buffer), and the provided 5X Reaction Buffer A supplemented with 50 µM DTT, 2 mM CaCl_2_ and 50 µM ATP. 

As reported by Promega, R59949 has an IC_50_ of 267 μM in this assay. Therefore, DGKA activity was measured in the presence of different DGKA inhibitors at a final concentration of 400 μM or an equal volume of DMSO as a control. An equal volume of water was used as a negative control (no enzyme reaction), while *E. coli*-purified DGKA was used as the positive control (DGKA *E. coli*). The luminescence of the 384-well white plate was measured with a Tecan Spark 10 M Multimode Plate Reader, and the data were collected as Excel files and analyzed by GraphPad Prism 9.0 software.

### 2.7. Reverse Transcription Polymerase Chain Reaction Assay

Total RNA was extracted from 2 × 10^6^ pelleted cells using the MagMAX^TM^ Viral/Pathogen II Nucleic Acid Isolation Kit (Thermo Fisher Scientific, Waltham, MA, USA) on a Kingfisher automated nucleic acid extractor (Thermo Fisher Scientific). To adapt the protocol to cell line samples, the pellets were dissolved in 200 μL of homogenization solution (Promega) before proceeding according to the manufacturer’s instructions.

The High-Capacity cDNA Reverse Transcription Kit (Thermo Fisher Scientific) was used to synthesize cDNA from 1 μg of RNA as per the prescribed protocol.

The relative expression of 5 isoforms of DGK (*DGKA* Assay ID Hs00176278_m1, *DGKD* Assay ID Hs01114141_m1, *DGKE* Assay ID Hs00177537_m1, *DGKG* Assay ID Hs00176315_m1, *DGKZ* Assay ID Hs05025727_m1) was assessed by quantitative real-time PCR (qRT-PCR) analysis using TaqMan primers, probe assays (Thermo Fisher Scientific) and TaqMan Fast Advanced Master Mix (Thermo Fisher Scientific); the amplification program was initial denaturation at 94 °C for 20 s, followed by 40 cycles of denaturation at 94 °C for 3 s and annealing/extension at 60 °C for 30 s. The qPCRs were performed in biological and technical triplicates.

Glucuronidase beta (*GUSB* Assay ID Hs00939627_m1) and glyceraldehyde-3-phosphate dehydrogenase (*GAPDH* Assay ID Hs02758991_g1) were used as internal reference genes. The fold changes were calculated by the 2−ΔΔCT method using the average expression of the two reference genes, and the relative expression was normalized using PBLs as a control (CFX Maestro software 1.1, Biorad, Hercules, CA, USA).

To verify the statistical significance of differences between the leukemic cell lines and PBLs, an independent sample t-test was conducted using Excel software (Microsoft, Redmond, Washington, DC, USA). The results were presented as mean ± SD. Differences with *p*-values of <0.05 were considered statistically significant.

### 2.8. AlamarBlue and Trypan Blue Viability Assays

The alamarBlue viability assay was performed to assess the effect of DGK inhibitors on cell viability. We tested the effects of the DGK inhibitors diluted in RPMI supplemented with 10% FBS at different concentrations ranging from 100 µM to 6.25 µM. For each sample, 50,000 cells per well were added to each well of 96-well plates in a volume of 100 μL, together with 10 μL of 10× inhibitor, and incubated at 37 °C, 5% CO_2_, for 24 h. All experiments included untreated cells as the viability control and Triton x100 treated cells as a cytotoxicity control and were conducted minimizing light exposure. After 24 h, alamarBlue (1.5 mg/mL) was kept at room temperature and diluted 1:10 with PBS 1X. Twelve μL of diluted alamarBlue were added to the wells, for a total volume of 122 μL. Successively, the plates were incubated for an additional 24 h at 37 °C. After 24 h, the fluorescence was measured with excitation and emission values of 560 nm and 590 nm, respectively, using a Tecan Spark 10 M Multimode Plate Reader.

Each experimental condition was performed in triplicate or quadruplicate, using cells treated with equal amounts of the solvent DMSO as a reference and medium as a background. Thus, for the dose–response curve fitting, fluorescence was normalized as:$$Viability\;\left(\%\right)=\frac{\left(\mathrm{sample}\;\mathrm{fluorescence}\;\mathrm{medium}\;\mathrm{fluorescence}\right)}{\left(\mathrm{mean}\;\mathrm{DMSO}\;\mathrm{control}\;\mathrm{fluorescence}\;\mathrm{medium}\;\mathrm{fluorescence}\right)}\times 100$$

Data are the results of at least 9 independent experiments analyzed with GraphPad Prism 7 as [inhibitor] vs. normalized response curve (variable slope, automatic outlier exclusion).

The trypan blue viability assay was instead performed on HL-60 and HEL cells to explore the effect of DGK inhibition on cancer cell number. The same plating protocol as the alamarBlue assay was adopted, i.e., 50,000 cells/well in a 96-well plate in triplicate in a volume of 100 μL with 10 μL of 10× inhibitor (R59022 or R59949) or the equivalent amount of DMSO (control DMSO) or complete growth medium (untreated control). After 48 h of incubation, the trypan blue reagent was added to the cell cultures (1:1 *v*/*v*), and the number of live/dead cells was automatically recorded by a Countless automated cell counter (Invitrogen, Waltham, MA, USA). Each experiment was performed three times, and results are reported as the number of total live/dead cells, graphed, and analyzed with GraphPad Prism 7 with a one-way ANOVA and paired *t*-test. 

### 2.9. Real-Time Viability Assay 

To evaluate the long-term effects of DGKA inhibitors, a viability assay was performed with an xCELLigence Real-Time Cell Analyzer (RTCA, Roche, Basel, Switzerland). The E16 plate impedance was measured every minute for the first 10 min to monitor the thermal balancing process and then every 15 min for 100 h to acquire the data. The 2× inhibitors were prepared in RPMI supplemented with 10% FCS, and 100 µL of these treatments were added to each E16 plate well. Next, the E16 plate was placed in the RTCA at 37 °C for 10 min for thermo-equilibration. Meanwhile, 1 million cells were resuspended in 2 mL of complete RPMI at a final concentration of 50,000 cells per 100 µL. After the plate equilibrated, 100 µL of cells were plated in each well. Finally, the plate was placed in the RTCA for data acquisition over a 100-hour period, and the impedance was measured by sensor electrodes located at the bottom of the plate. Subsequently, the plate was observed under an optical microscope to verify the well’s appearance and the presence or absence of precipitated crystals. The E16 plate was regenerated by washing twice with 1X PBS, followed by 100 µL trypsin per well and 2 additional washings with 1X PBS, according to the procedure described in Stefanowicz-Hajduk et al. [29]. Each E16 plate was reused three times. 

Cell index data at 91 h were analyzed by GraphPad, performing one-way ANOVA and a paired *t*-test to determine statistical significance. 

### 2.10. Cell Cycle

The effects of DGK inhibitors on the cell cycle were evaluated by flow cytometry using a DraQ5 stain (Thermo Fisher Scientific). HL-60 cells were treated with R59022 at 25 µM or R59949 at 100 µM. The concentrations were chosen based on the results of the alamarBlue viability assays. All treatments were diluted in RPMI with 10% FBS. A total of 250,000 cells per well were plated in 500 μL of medium (24-well plate) with 50 μL of 10× treatment solution. Cells incubated with a complete medium and DMSO 1% were used as a viability-positive control. The plates were then incubated at 37 °C with 5% CO_2_. After 24 h, cells were counted with trypan blue (Thermo Fisher Scientific) to equalize the cell number in each sample before staining. Subsequently, 120,000 cells were stained with DraQ5 at a concentration of 10 μM and diluted in 1X PBS for 30 min in the dark at room temperature before analysis. Samples were analyzed on a FACSCanto II using BD FACSDiva software. Data were gated based on forward light scatter area (FSC-A) vs. sideward light scatter area (SSC-A) to exclude debris and FSC-A vs. FSC height (FSC-H) to exclude cell doublets. Post-acquisition analysis was performed using FlowJo^®^ 7.6.3 to identify cell cycle phases.

## 3. Results

### 3.1. Overexpression of Selected DGK Isoforms in AML 

To gain insights into the role of DGK in tumors, we interrogated the GEPIA2 [25] multiple gene analysis tool using the 10 DGK isoforms as input and compared their expression profiles in the TCGA database with those in matched normal tissue samples from the TCGA and GTEx databases. We found that several DGK isoforms were overexpressed in acute myeloid leukemia (LAML) in comparison with bone marrow (Figure 1A). In particular, significant overexpression was observed for type I *DGKA* and *DGKG*, type II *DGKD*, type III *DGKE* and type IV *DGKZ* (Figure 1B). Conversely, *DGKH* and *DGKQ* showed no significant alteration, while *DGKB*, *DGKK* and *DGKI* were expressed at low levels and thus not further investigated. Proteomic analysis of a subset of TCGA LAML samples using label-free technology only yielded quantifiable results for DGKA and DGKZ proteins [30]. Even though the number of patients and controls was insufficient to establish statistical significance at the protein level, we observed higher levels of DGKA in several AML samples compared to CD34^+^ or lin^−^ controls, whereas no such difference was seen for DGKZ (Appendix A). Thus, the TCGA data indicate that DGK activity is relevant in AML biology, particularly highlighting the significance of DGKA in this disease. 

To further investigate the functional relevance of DGK isoforms in AML transformation, we turned to the independent BeatAML database [26], which includes two sets of reference samples from healthy donors, namely CD34^+^ cells and mononucleated cells from BM. Using healthy donor CD34^+^ cells as a reference, we detected overexpression of *DGKA*, *DGKG*, *DGKD* and *DGKQ* in tumor samples, while *DGKH* and *DGKZ* were unchanged. *DGKE* showed a slight decrease in expression, while *DGKB*, *DGKK* and *DGKI* were expressed at low or undetectable levels. These trends are essentially in line with the TCGA results, except for the lack of evidence for increased expression of *DGKE* and *DGKZ* in tumor samples. Of note, in contrast with the TCGA findings, the BeatAML database revealed that BM cells express all DGK isoforms at very high levels, often surpassing those observed in tumor samples (Appendix A). This may suggest a tendency for increased DGK expression upon differentiation, highlighting the challenge of establishing the appropriate “control cell type” for a heterogeneous disease such as AML.

To explore the functional implications of DGK isoform expression in AML transformation, we examined the correlation between their expression levels and overall survival in AML patients from the TGCA database. When comparing the survival rate of high expressors (red) with that of low expressors (blue) or unchanged (green), we observed a decreased survival in patients with high levels of *DGKA*. Interestingly, high expression of *DGKG*, a class I DGK isoform that is overexpressed, similar to *DGKA*, showed an inverse correlation with prolonged survival. *DGKE, DGKD* and *DGKZ*, despite being overexpressed in AML, did not show a significant correlation with survival, similar to the non-overexpressed *DGKH* and *DGKQ* isoforms (Figure 2). However, due to the small number of patients and high heterogeneity of the disease, these data should be interpreted with caution. Indeed, when we analyzed patients from two other independent databases using the same approach, we obtained somewhat different results. In the BeatAML database, we observed a significant correlation between high *DGKH* expression and decreased survival (Appendix A). Likewise, in the TARGET database, which primarily includes pediatric patients with higher survival rates, high *DGKH* expression is negatively correlated with survival, and aberrant levels of *DGKQ* are correlated with decreased survival (Appendix A).

When examining the relationship between the expression of DGK family members and the molecular subtypes of AML in the BeatAML dataset [26], we observed that the expression patterns of DGK isoforms were relatively similar across all subtypes. *DGKH* consistently showed the lowest expression except in acute monocytic leukemia (AMOL), while *DGKZ*, *DGKD*, *DGKQ* and *DGKA* exhibited the highest expression levels (Appendix A). In good agreement, we did not observe a significant and reproducible enrichment of any isoform in a particular AML subtype when analyzing three different databases (i.e., TCGA, BeatAML and Clinseq) [31]. Thus, the analyses of the TCGA, BeatAML and TARGET databases demonstrate a robust expression of DGK family members in AML without a clear association with a specific molecular subtype. Although the prognostic significance of such an expression profile remains to be fully elucidated, some studies have revealed a negative association between *DGKA*, *DGKH* and *DGKQ* expression and survival outcomes.

To gain a more complete picture of the functional roles of individual DGK isoforms, we analyzed the co-expression of genes associated with each isoform using data from the BeatAML study [26] through the CBioPortal. Specifically, we performed hierarchical clustering of the isoforms based on their co-expressed genes (Figure 3). Notably, *DGKD*, the most expressed isoform, showed an association with a specific gene program characterized by upregulation of intracellular vesicular membrane trafficking/autophagy and receptor signaling, coupled with the downregulation of genes associated with DNA/RNA metabolism. This gene program was only partially shared with *DGKA*, which was rather associated with upregulated genes related to signaling in the immune system. On the other hand, *DGKE* was associated with the downregulation of genes related to membrane trafficking and organelle assembly and the upregulation of genes related to transcription and protein synthesis. *DGKG* constituted a separate branch that encompassed upregulated genes regulating the metabolism of carbohydrates and neutrophil activity, partly shared with *DGKZ*, which is co-expressed with genes related to not only neutrophil activity but also cytoskeletal remodeling and vesicular trafficking (Figure 3A,B).

Overall, these findings support the idea of isoform-specific functionalities for DGKs, revealing a connection between *DGKA* and *DGKG* and distinct genetic programs that are consistent with their divergent effects on patients’ survival. In addition, it seems that DGK isoforms share functions related to the control of cellular signaling pathways involved in vesicular trafficking and possibly differentiation, which aligns with their expected roles in regulating membrane composition.

To explore whether the upregulated and downregulated genes corresponded to specific cell types, we investigated their association with different AML blasts along the leukocyte differentiation pathways (Figure 3C,D). We found that *DGKA* is associated with gene profiles typical of the thymus, T-blast (MOLT4) spleen and blood, while *DGKG* is associated with a more stem cell-like gene program found in BM and CD34^+^ cells.

### 3.2. Effect of DGK Inhibitors on AML Cell Lines

As our data suggested an alteration of diacylglycerol signaling in AML and an isoform-specific contribution of DGK to cell transformation, we sought to determine the potential of DGK inhibitors as therapeutic targets in AML. For this purpose, we evaluated their cytotoxic effects on the AML cell lines HL-60 (promyelocytic leukemia) and HEL (erythroleukemia). In parallel, we used activated PBLs from healthy donors as references, as they express high levels of DGKs that are known to play a role in proliferation and survival [32]. Interestingly, HL-60 and HEL cell lines exhibited low *DGKA* and high *DGKZ* expression levels compared to PBLs. The *DGKG* content was lower in HEL but higher in HL-60 (Appendix A). We tested various DGK inhibitors, including commercially available inhibitors targeting DGKA and other isoforms as well as serotonin receptors (R59022 and R59959) [33], a DGKA-specific ATP competitive inhibitor (CU-3) [34], and an in-house-developed DGKA-specific inhibitor that does not affect serotonin receptors (AMB639752) [22,23]. All tested molecules showed activity at a concentration of 400 μM against purified DGKA in an in vitro kinase assay (Appendix A). 

We initially evaluated the acute effects on cell viability using the alamarBlue assay after a short exposure (24 h) to drug concentrations ranging from 6.25 to 100 μM. Interestingly, the DGKA-specific inhibitors AMB639752 and CU-3 showed limited efficacy in HL-60 cells, while R59949 decreased viability with a half maximal inhibitory concentration (IC_50_) of 72 ± 2 μM. R59022 emerged as the most effective molecule with an IC_50_ of 32 ± 2 μM (Figure 4). 

Viability tests using trypan blue on HL-60 cells (Appendix A) revealed that both untreated and DMSO control cells were proliferating. Nevertheless, the administration of the lowest dose of R59022 (50 µM) exerted a cytostatic effect in HL-60 cells, whose number did not increase over time. Moreover, the highest dose (100 µM) was particularly cytotoxic for this cell line, leading to a considerable decrease in viable cells. In the case of R59949, both doses were almost cytostatic, inhibiting cell proliferation over time. It is worth noting that since the count was performed 48 h after incubation with the inhibitors and given that the automated cell counter is not able to detect fragments and debris derived from cells dead before this time point, the number of dead cells might have been underestimated. 

To assess the long-term effects of the DGK inhibitors on cell viability, we conducted real-time monitoring for over 100 h using the xCELLigence Real-Time Cell Analyzer at a medium-low 50 μM drug concentration. Interestingly, we observed that R59022 at this concentration effectively halted cell growth and induced cell death within the first day of treatment. By the second day, cell viability had decreased to background levels (Figure 5). In contrast, both CU-3 and AMB639752 did not significantly affect cell proliferation even after prolonged treatment or at a higher concentration of 100 μM (Figure 5). It should be pointed out that R59949 was not suitable for use with this technique due to the formation of small crystals upon prolonged incubation, which interfered with the impedance-based measurement of cell viability.

We further investigated the effect of R59022 and R59949 on the cell cycle of HL-60 cells. After a 24-hour incubation with inhibitors at doses that did not completely kill the cell population (i.e., R59022 25 µM, R59949 100 µM), we used the DraQ5 probe to assess the DNA content. We observed a reduction in the G1 phase paralleled by an increase in the S phase with both inhibitors, suggesting a blockage in the S phase in addition to the reported toxicity (Figure 6).

Similar results were obtained in HEL cells using the alamarBlue assay. The DGKA-specific inhibitors AMB639752 and CU-3 were poorly effective, while R59949 decreased viability with an IC_50_ of 80 ± 1 μM. R59022 was the most effective inhibitor, with an IC_50_ of 49 ± 2 μM (Figure 4). In line with these findings, the trypan blue live/dead cell count (Appendix A) revealed once more that DMSO alone could not significantly affect cell proliferation. Nonetheless, treatment with R59022 at 50 µM led to a slight cytotoxic effect, which was much more pronounced at a concentration of 100 µM. A similar trend was observed with R59949 in this cell line. 

In contrast, cultured PBLs used as a reference, primarily consisting of CD8^+^ T cells, displayed a different inhibitor sensitivity profile. The DGKA-specific inhibitors AMB639752 and CU-3 showed activity with IC_50_ values of approximately 100 μM and 67 ± 3 μM, respectively. R59022 also showed growth inhibitory activity with an IC_50_ of 43 ± 2 μM, while R59949 was the most effective molecule with an IC_50_ of 22 ± 2 μM (Figure 4). 

Taken together, these results suggest selective sensitivity of HL-60 and HEL to broad-spectrum DGK inhibitors (i.e., R59022, R59949), while DGKA-specific inhibitors, such as CU-3 and AMB639752, appear to be ineffective at concentrations that are capable of reducing PBL viability, as expected. These findings indicate that DGK activity is relevant in AML, but inhibiting the DGKA isoform alone is not sufficient to decrease cell survival, at least in these models. 

## 4. Discussion

Despite their established role in T cell biology, the role of DGK isoforms in leukemogenesis remains largely unexplored. To assess their involvement in tumorigenesis, here we have analyzed the TCGA and BeatAML datasets and found that several DGK isoforms are overexpressed in AML, particularly *DGKA, DGKG* and *DGKD* (Figure 1 and Appendix A). High *DGKA* expression in the TCGA dataset negatively correlates with survival, whereas high *DGKG* expression shows the opposite trend (Figure 3). In the BeatAML and TARGET datasets, high *DGKH* expression is associated with shorter survival. These differences may be due to variations in patient populations across the dataset, with BeatAML being more recent than TGCA and TARGET rather than focusing on pediatric AML patients. Although these data may imply a role for *DGKA* and *DGKH* in AML pathogenesis, they should be interpreted with the utmost caution, as it is challenging to identify a suitable “healthy reference tissue” for a disease like AML, which affects cell differentiation. Furthermore, the correlation with survival can vary between different datasets (Appendix A). 

In line with the prognostic value of DGK expression in AML, Li et al. recently included *DGKA* in a six-gene risk signature related to lipid metabolism and demonstrated its utility for AML prognosis [35]. Moreover, among the several rare translocations observed in AML patients, a ZNF273–DGKA translocation has been recently reported [36]. This translocation results in a fusion protein where the regulatory N-terminal domain of DGKA is replaced by the entire ZNF273 Zn finger domain, potentially leading to an active membrane-localized DGKA. Similarly, a ZFAND3–DGKH translocation, retaining the catalytic DGK domain, was documented in Philadelphia-like acute lymphoblastic leukemia [37]. Furthermore, we have recently reported nuclear localization of DGKA in the human erythroleukemia cell line K562 and a cytostatic effect of DGK inhibitors that impairs cell cycle progression [38]. Collectively, these data suggest that DGKs are involved in the differentiation and cell cycle progression of leukemia cells in an isoform-specific fashion and point to DGKs as attractive therapeutic targets.

The human promyelocytic leukemia cell line HL-60, known to differentiate toward the monocytic lineage in response to diacylglycerol analogs, has been widely used to study the role of DGK in myeloid differentiation and AML pathogenesis. A previous report highlighted the importance of *DGKZ* for HL-60 viability, as knockdown of this isoform inhibits proliferation, induces cell cycle arrest at the G2/M phase, and promotes apoptosis, possibly through a MAPK/survivin/caspase pathway [24]. Our study also demonstrates impaired cell cycle progression upon treatment with R59022 or R59949 in HL-60 cells (Figure 6). In the undifferentiated state of HL-60 cells, multiple DGK isoforms, including *DGKD*, *DGKE, DGKG* and *DGKZ*, are expressed, while the *DGKA* isoform is upregulated specifically during differentiation toward the granulocyte lineage [39]. Conversely, forced expression of *DGKG* negatively affects HL-60 differentiation toward the macrophage lineage [40]. These findings are in good agreement with our observation of different transcriptional programs associated with each DGK isoform and the negative predictive value of high *DGKA* expression compared to the positive effect of high *DGKG* (Figure 3). 

The question of how two isoforms, which share a similar structure and catalytic activity, can exert such divergent effects is intriguing. The precise expression level of a gene is determined by a complex network of transcriptional, epigenetic and post-transcriptional regulators that operate in a composite and combinatorial way in context-specific regulatory loops. Transcription factors and microRNAs are recognized as key players in the regulation of gene expression; they are often operate in mixed regulatory loops and frequently dysregulated in disease contexts, including cancer. Thus, understanding the main transcriptional and post-transcriptional regulators that underlie the distinct expression programs specific to each DGK isoform represents an important future endeavor. 

Our findings support the involvement of DGKs in leukemic transformation, as evidenced by the decrease in HL-60 and HEL cell viability, impairment of the cell cycle, and induction of cell death by treatment with R59922 or R59022. However, the lack of effectiveness of the highly DGKA-specific inhibitors CU-3 and AMB639752 on these cell lines implies that targeting this single isoform may not be sufficient to decrease cell viability in myeloid cell lines. This is in stark contrast with cultured PBLs, which are enriched in CD8^+^ T cells and rely on DGKA for IL-2-driven proliferation [41]. It remains unclear whether the efficacy of R59022 and R59949 on leukemia cells is due to off-target effects, such as those reported on serotonin receptors [33], or to a broader activity on other DGK isoforms at the relatively high concentrations used in this study. In this regard, ritanserin, a serotonin receptor antagonist similar to R59949 and R59022, has been used in several clinical trials at doses supposed to inhibit DGKA [33,42], suggesting its potential for drug repurposing in AML.

The signaling pathways affected by DGK isoforms overexpressed in AML are still unknown and likely to be isoform-specific. Of note, among the diacylglycerol-dependent PKC, the epsilon isoform has emerged for its pro-oncogenic function in solid cancers and its role in hematological cell differentiation. Interestingly, *PRKCE* is upregulated in AML, where it is associated with poor outcomes. Mechanistically, PKCepsilon plays an antiapoptotic and antioxidative role and promotes chemoresistance through an increase in P-glycoprotein activity [43,44]. This suggests that the protective role of DGKG may also involve the removal of cellular DAG in addition to its reported role in modulating glucose import in cancer cells [45]. Conversely, DGKA and DGKZ may synergistically modulate cell proliferation by affecting phosphatidic acid-sensitive molecules, such as atypical PKC [46] and mTOR [47], as well as by controlling blast differentiation along different lineages [39,40].

## 5. Conclusions

In conclusion, our findings indicate frequent overexpression of DGK isoforms in AML patients compared to healthy donor cells. Interestingly, high *DGKA* expression is associated with a poor prognosis, while increased *DGKG* expression is linked to a better prognosis, although these associations seem to vary across different datasets. Importantly, we show that selective inhibition of DGKA alone is ineffective in killing AML cell lines, while inhibitors with broader specificity against DGK isoforms show efficacy in reducing the viability and inducing cell cycle arrest in these cells. Thus, our results highlight the need for further investigation into the specific biological roles of DGK isoforms in AML and their potential as therapeutic targets.

## Figures and Tables

**Figure 1 biomedicines-11-01877-f001:**
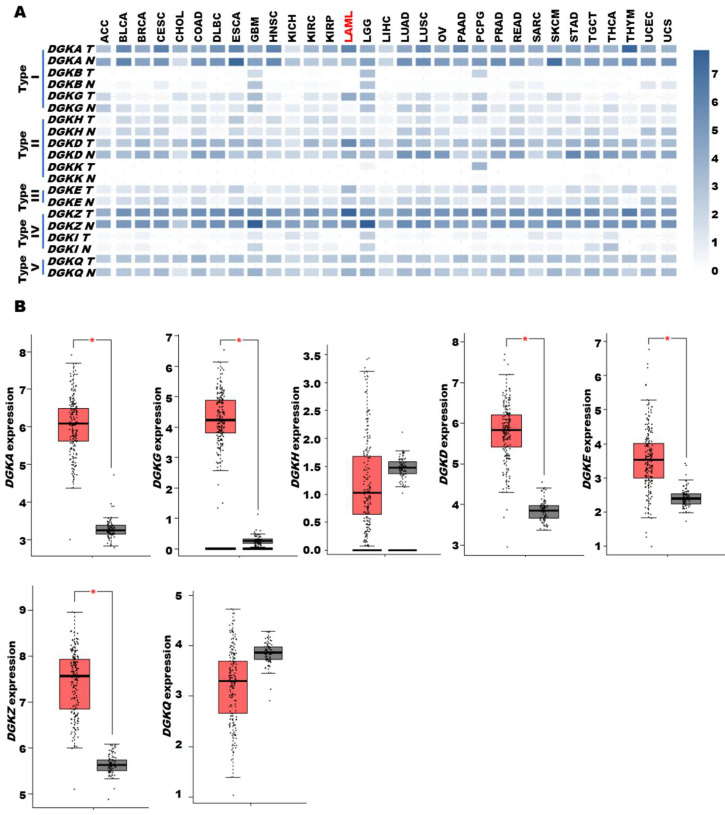
DGK expression in AML based on data from the TCGA database. TCGA tumor data are shown as log2 compared to TCGA and GTEx normal tissue data. (**A**) DGK family expression in the TCGA dataset. (**B**) DGK family expression in AML. The dataset comprised 173 tumor samples (red) and 70 normal tissues (grey). A single asterisk indicates a *p*-value of ≤0.01. TCGA cancer name details: adrenocortical carcinoma (ACC); bladder urothelial carcinoma (BLCA); breast invasive carcinoma (BRCA); cervical squamous cell carcinoma and endocervical adenocarcinoma (CESC); cholangio carcinoma (CHOL); colon adenocarcinoma (COAD); lymphoid neoplasm diffuse large B-cell lymphoma (DLBC); esophageal carcinoma (ESCA); glioblastoma multiforme (GBM); head and neck squamous cell carcinoma (HNSC); kidney chromophobe (KICH); kidney renal clear cell carcinoma (KIRC); kidney renal papillary cell carcinoma (KIRP); acute myeloid leukemia (LAML); brain lower grade glioma (LGG); liver hepatocellular carcinoma (LIHC); lung adenocarcinoma (LUAD); lung squamous cell carcinoma (LUSC); mesothelioma (MESO); ovarian serous cystadenocarcinoma (OV); pancreatic adenocarcinoma (PAAD); pheochromocytoma and paraganglioma (PCPG); prostate adenocarcinoma (PRAD); rectum adenocarcinoma (READ); sarcoma (SARC); skin cutaneous melanoma (SKCM); stomach adenocarcinoma (STAD); testicular germ cell tumors (TGCT); thyroid carcinoma (THCA); thymoma (THYM); uterine corpus endometrial carcinoma (UCEC); uterine carcinosarcoma (UCS); uveal melanoma (UVM).

**Figure 2 biomedicines-11-01877-f002:**
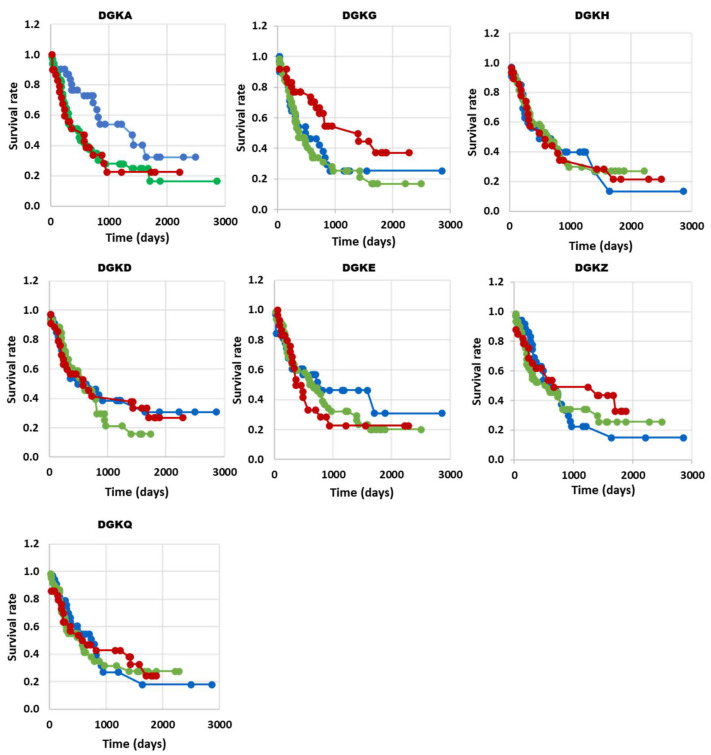
DGK expression and survival correlation in AML (TGCA database). The high, low and normal DGK groups are shown in red, blue and green, respectively. *DGKA* n (up) = 26, n (no change) = 50, n (down) = 30, with *p* = 0.04 *; *DGKG* n (up) = 30, n (no change) = 48, n (down) = 27, with *p* = 0.03 *; *DGKH* n (up) = 25, n (no change) = 50, n (down) = 29, with *p* = 0.99; *DGKD* n (up) = 28, n (no change) = 47, n (down) = 31, with *p* = 0.76; *DGKE* n (up) = 28, n (no change) = 53, n (down) = 26, with *p* = 0.51; *DGKZ* n (up) = 23, n (no change) = 45, n (down) = 31, with *p* = 0.55; *DGKQ* n (up) = 28, n (no change) = 47, n (down) = 29, with *p* = 0.92; * *p* < 0.05.

**Figure 3 biomedicines-11-01877-f003:**
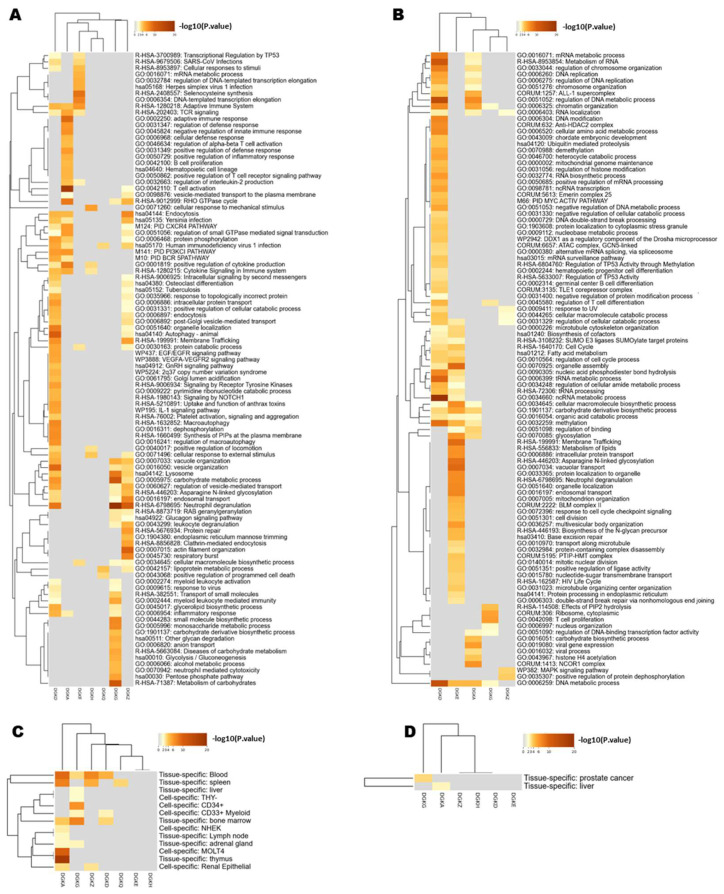
Analysis of DGK co-expressed genes in the BeatAML database. (**A**) Heatmap representing the hierarchical clustering of the top enriched pathways obtained by Metascape, using as input the set of genes positively correlated (q-value < 0.05; Spearman’s correlation > 0.40) with the DGK isoforms (columns), according to cBioPortal data. Each row represents a functional cluster. (**B**) Heatmap representing the hierarchical clustering of the top enriched pathways obtained by Metascape, using as input the set of genes negatively correlated (q-value < 0.05; Spearman’s correlation < −0.40) with the DGK isoforms (columns), according to cBioPortal data. Each row represents a functional cluster. (**C**) Heatmap of PaGenBase enrichment for genes co-expressed with DGK isoforms from Metascape, obtained giving as input the genes positively correlated with the DGK isoforms (q-value < 0.05; Spearman’s correlation > 0.40). Each row represents tissues and cell types. (**D**) Heatmap of PaGenBase enrichment for genes co-expressed with DGKs from Metascape, obtained by giving as input the genes negatively correlated with the DGK isoforms (q-value < 0.05; Spearman’s correlation < −0.40). Each row represents tissues and cell types.

**Figure 4 biomedicines-11-01877-f004:**
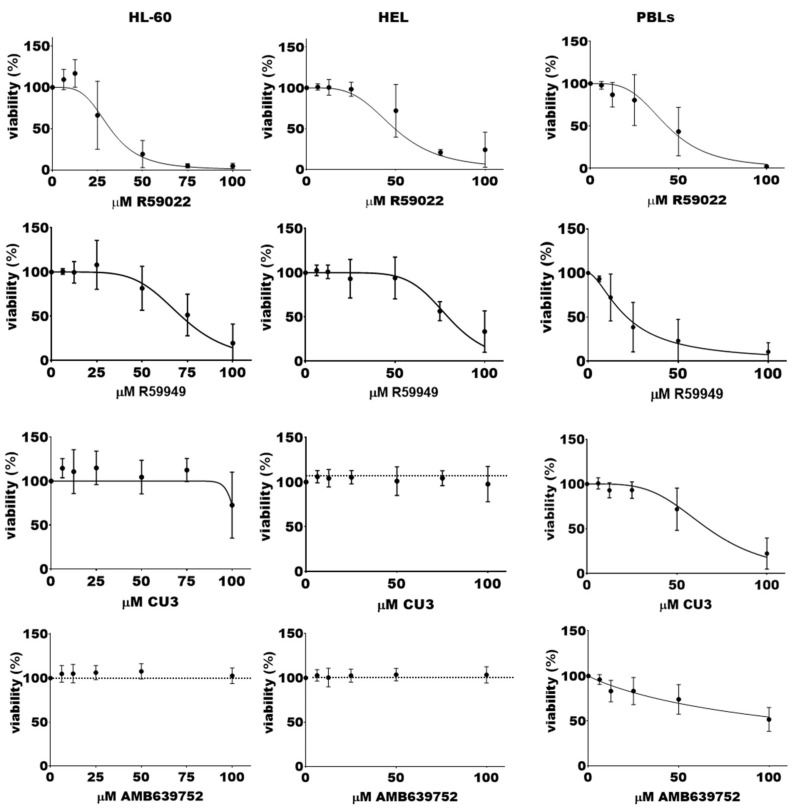
Effects of DGK inhibitors on AML cell lines viability. A total of 50,000 cells were treated with increasing concentrations of the indicated inhibitor or vehicle for 24 h, followed by an alamarBlue viability assay for an additional 24 h. Data are presented as the mean ± SEM of at least nine independent experiments run in triplicates or quadruplicates, plotted as [inhibitor] vs. % viability.

**Figure 5 biomedicines-11-01877-f005:**
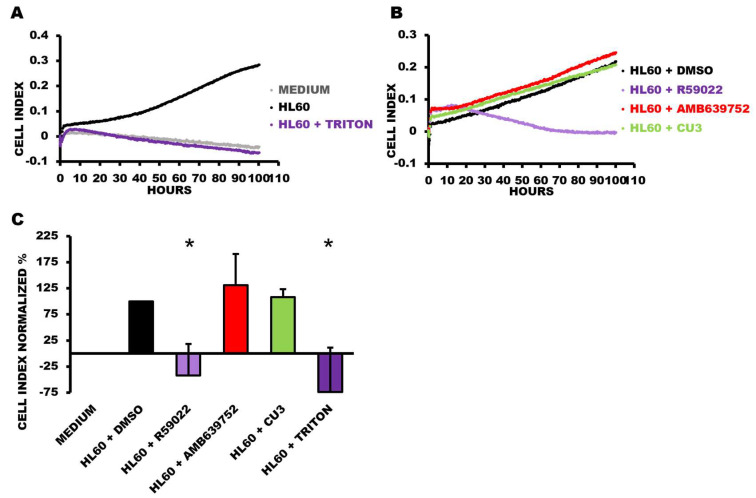
Viability time course of HL-60 cells treated with DGKA inhibitors. HL-60 cells were plated in the presence or absence of the indicated treatments at a concentration of 50 µM and followed for 100 h. Data are presented as an impedance trend over time. Representative experiment showing (**A**) control conditions and (**B**) the treated samples. (**C**) The graph represents the mean of four independent experiments at 91 h. The data were normalized to the respective control (HL60 + DMSO) and shown as the mean ± SD of the percentage. * *p* < 0.05, *t*-test vs. HL60 + DMSO.

**Figure 6 biomedicines-11-01877-f006:**
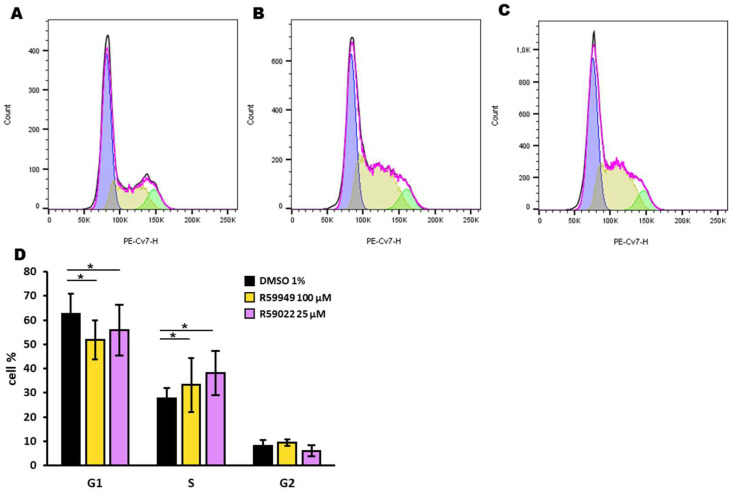
Effect of DGK inhibitors on the HL-60 cell cycle. HL-60 cells were treated with DMSO (1%), R59022 (25 µM), or R59949 (100 µM) and incubated for 24 h before DraQ5 staining. Black line: experimentally observed data; purple line: data estimated by the analysis model; blue: estimated G1 phase; yellow: estimated S phase; green: estimated G2 phase. Representative experiment indicating cells treated with (**A**) DMSO, (**B**) R59022, or (**C**) R5949. (**D**) Percentage of cells in each phase of the cell cycle for each treatment. Data are expressed as the mean ± SD of five independent experiments. * *t*-test, *p* < 0.05.

## Data Availability

Not applicable.

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
