# Peer review of "Role of Diacylglycerol Kinases in Acute Myeloid Leukemia"

_biomedicines, 2023, doi:10.3390/biomedicines11071877_

Round 1

Reviewer 1 Report

This manuscript has examined DGK isoform expression in AML and includes some preliminary data on potential isoform-specific roles of several DGKs. Overall, the manuscript is interesting and has provided new data that can be used as a basis for further work. My major comment is that I feel that overall conclusions have been overstated (see below). Figures are well-presented and referencing is appropriate, with a good selection of relevant publications included. There are numerous English language errors encompassing both grammatical and syntax errors; these will require correction in order that the manuscript can be properly reviewed. The authors could consider the following comments and corrections.

1. In this study, it is noteworthy that associations between DGK isoform expression and survival of AML patients that the authors identified using the TCGA database could not be reproduced using data from the Beat-AML database. As acknowledged by the authors, these databases contain information from only small numbers of patients. However, I feel that due to both the irreproducibility of findings and the small numbers of patients that the results cannot be presented as findings of the study (e.g. in the Abstract or Conclusions) without qualifying statements (caveats). The absence of consistent supporting evidence from cell line studies appears to reinforce that there are no clear associations between expression levels of individual DGK isoforms and AML outcomes.

2. Lines 353-355 (These data reinforce the notion of….): I feel that this sentence overstates / overinterprets results as these conclusions rely on bioinformatics data only (results have not been verified with functional studies), and in addition, the authors have already shown that at least part of the data is not reproducible (see comment above).

3. Do DAG kinase inhibitor cause cell death or are their effects principally growth (cell cycle) arrest? Because data are normalised against DMSO controls, it is difficult to differentiate these outcomes.

4. Line 152: Please state the source of the GST-DGKA construct.

5. Line 173: The appropriate reference is missing (“as previously described”).

6. The authors have used t-tests throughout their study (e.g. lines 244-245). Have they checked that this is the most appropriate statistical test for their data (are data normally distributed)? Would non-parametric tests be more appropriate?

7. Figure 1 legend: Please add an explanation for the asterisk in the figures (presumable this relates to statistical significance). The font size in lines 314-327 should be made consistent with the figure legend font size.

8. Lines 335-336: This sentence is incomplete.

9. Reference 24 is referred to throughout the text as either Taynor or Tyner (Tyner is correct). This requires correction (this type of error may not be picked up by an English language editor).

10. The legend for Figure 3 requires amendment. The panels are labelled A, B, C, D but the legend includes A, B, C only. (I think that panel C has been split and that C and D should be joined into a single panel).

11. Supplementary Figure 2: Please state the numbers of healthy donor CD34+ and healthy donor Lin- cases used in the graphs. It seems as though there may be only 3 cases in each? The graph presentation should also be described (is it median/mean +/- what measurement?). Is this sort of presentation legitimate for the number of cases in each of the healthy donor groups? (The word ‘healthy’ is also spelled incorrectly in each of the graphs).

12. Supplementary Figure 3: Please add the number of samples in each category and an explanation of abbreviations to the figure legend.

There are numerous minor English language errors throughout the manuscript which I feel should be corrected before it is re-reviewed. While most of the errors are simple grammatical or syntax errors that can be easily identified and corrected by an English language editor (a person, not a programme), there are several places where incorrect scientific terminology is used. These errors may not be detected by an English language editor who is not familiar with terminology commonly used in this field of research, but would be easily recognised by a scientific reviewer.

Author Response

This manuscript has examined DGK isoform expression in AML and includes some preliminary data on potential isoform-specific roles of several DGKs. Overall, the manuscript is interesting and has provided new data that can be used as a basis for further work. My major comment is that I feel that overall conclusions have been overstated (see below). Figures are well-presented and referencing is appropriate, with a good selection of relevant publications included. There are numerous English language errors encompassing both grammatical and syntax errors; these will require correction in order that the manuscript can be properly reviewed. The authors could consider the following comments and corrections.

We thank reviewer 1 for appreciation of the manuscript, which underwent professional editing (certificate in attachment). As those editing deeply changed the phrasing, we are uploading the new version with accepted changes for clarity. A comparison with tracked changes is in attachment for your reference.

  1. In this study, it is noteworthy that associations between DGK isoform expression and survival of AML patients that the authors identified using the TCGA database could not be reproduced using data from the Beat-AML database. As acknowledged by the authors, these databases contain information from only small numbers of patients. However, I feel that due to both the irreproducibility of findings and the small numbers of patients that the results cannot be presented as findings of the study (e.g. in the Abstract or Conclusions) without qualifying statements (caveats). The absence of consistent supporting evidence from cell line studies appears to reinforce that there are no clear associations between expression levels of individual DGK isoforms and AML outcomes.

In order to address this issue, we analyzed a further dataset (TARGET, containing mainly pediatric AML) and reanalyzed the data using a standardized approach through the IPA (QUIAGEN) tool. The results that are presented in the revised version of the paper confirm the initial finding of a significant association of DGKA with shorter survival and DGKG with prolonged survival in the TGCA dataset, and a difference with BeatAML and TARGET studies were instead high DGKH correlates with poorer prognosis. The discussion of the possible bases of such discrepancies is now present in the revised manuscript.

  1. Lines 353-355 (These data reinforce the notion of….): I feel that this sentence overstates / overinterprets results as these conclusions rely on bioinformatics data only (results have not been verified with functional studies), and in addition, the authors have already shown that at least part of the data is not reproducible (see comment above).

As for the previous point we understand that those data are based on retrospective analysis of datasets, but we think database exploration and comparisons are really relevant for further functional studies. The caveats associated with those analysis are better presented in the discussion.  

  1. Do DAG kinase inhibitor cause cell death or are their effects principally growth (cell cycle) arrest? Because data are normalised against DMSO controls, it is difficult to differentiate these outcomes.

We fully agree with the necessity of better clarifying this issue, thus we carried out further experiments with R59022 and R59949. The trypan blue assay allows counting cells at the end of the experiment and demonstrates a mainly cytostatic effect at 50μM and a cytotoxic effect at 100μM in both HL-60 and HEL AML cell lines. A representative experiment is shown in supplementary figure 8 and results are discussed in the text.

  1. Line 152: Please state the source of the GST-DGKA construct.

“Baldanzi G, Cutrupi S, Chianale F, Gnocchi V, Rainero E, Porporato P, et al. Diacylglycerol kinase-alpha phosphorylation by src on Y335 is required for activation, membrane recruitment and hgf-induced cell motility. Oncogene 2008” is now referenced.

  1. Line 173: The appropriate reference is missing (“as previously described”).

Amended: “as described in section 2.5”.

  1. The authors have used t-tests throughout their study (e.g. lines 244-245). Have they checked that this is the most appropriate statistical test for their data (are data normally distributed)? Would non-parametric tests be more appropriate?

We thank the reviewer 1 for this note. Data in figure 1, 2, 3, supplementary 4, supplementary 5 and supplementary 6 were analyzed with validated bioinformatic tools as referenced. Experimental data interpolation in figure 4 was done using GraphPad Prism 7 as [inhibitor] vs. normalized response curve (variable slope, automatic outlier exclusion).  

Data in Fig. 5C, 6D, supplementary 1, supplementary 3, supplementary 7 and supplementary 8 were analyzed as one-way ANOVA with T test per multiple comparisons using GraphPad Prism 7. As usually done in the field we routinely carry out three-four independent replicas for each experiment. As suggested from the reviewer a non-parametric test may be more appropriate for those small sample sizes, we performed Mann-Whitney Test for Figure 5c and supplementary figure 8. As you can see in the following table the p value is still significant although values differ.

Figure 5c

Comparison

p value

T test

p value

Mann-Whitney Test

HL60+DMSO vs HL60+R59022

0.0268

0.0139

HL60+DMSO vs HL60+TRITON

0.0382

0.0139

Supplementary Figure 8

HL60 R59022

Comparison

p value

One way ANOVA (gaussian distribution)

p value

Mann-Whitney Test

DMSO vs R59022 50 µM

<0.0001

0.0495

DMSO vs R59022 100 µM

<0.0001

0.0495

HL60 R59949

Comparison

p value

One way ANOVA (gaussian distribution)

p value

Mann-Whitney Test

DMSO vs R59949 50 µM

<0.0001

0.0463

DMSO vs R59949 100 µM

<0.0001

0.0463

HEL R59022

Comparison

p value

One way ANOVA (gaussian distribution)

p value

Mann-Whitney Test

DMSO vs R59022 50 µM

<0.0001

0.0369

DMSO vs R59022 100 µM

<0.0001

0.0463

HEL R59949

Comparison

p value

One way ANOVA (gaussian distribution)

p value

Mann-Whitney Test

DMSO vs R59949 50 µM

<0.0001

0.0495

DMSO vs R59949 100 µM

<0.0001

0.0495

  1. Figure 1 legend: Please add an explanation for the asterisk in the figures (presumable this relates to statistical significance). The font size in lines 314-327 should be made consistent with the figure legend font size.

Done accordingly “A single asterisk indicates a p-value of ≤ 0.01.”.

  1. Lines 335-336: This sentence is incomplete.

Whole text was carefully revised.

  1. Reference 24 is referred to throughout the text as either Taynor or Tyner (Tyner is correct). This requires correction (this type of error may not be picked up by an English language editor).

The dataset is now referenced as BeatAML all around the manuscript and references were updated accordingly.

  1. The legend for Figure 3 requires amendment. The panels are labelled A, B, C, D but the legend includes A, B, C only. (I think that panel C has been split and that C and D should be joined into a single panel).

Indeed, a formatting error occurred, and part of the legend was missing. Everting is corrected in this revised version.

  1. Supplementary Figure 2: Please state the numbers of healthy donor CD34+ and healthy donor Lin- cases used in the graphs. It seems as though there may be only 3 cases in each? The graph presentation should also be described (is it median/mean +/- what measurement?). Is this sort of presentation legitimate for the number of cases in each of the healthy donor groups? (The word ‘healthy’ is also spelled incorrectly in each of the graphs).

New legend “Data from BeatAML database [26] comprising 671 tumor, 16 healthy pooled CD34+ and 18 healthy individual BM MNC samples, are shown as Normalized_RPKM, mean ± SD. Tumor sample and healthy pooled CD34+ cells are compared using one-way ANOVA, p<0.05 *, <0.01 **, 0.001***, 0.0001 ****. RPKM: Reads Per Kilobase per Million”.

  1. Supplementary Figure 3: Please add the number of samples in each category and an explanation of abbreviations to the figure legend.

Figure and legend revised (now supplementary figure 4).

Comments on the Quality of English Language

There are numerous minor English language errors throughout the manuscript which I feel should be corrected before it is re-reviewed. While most of the errors are simple grammatical or syntax errors that can be easily identified and corrected by an English language editor (a person, not a programme), there are several places where incorrect scientific terminology is used. These errors may not be detected by an English language editor who is not familiar with terminology commonly used in this field of research, but would be easily recognised by a scientific reviewer.

A professional English editing serviced was used to revise the manuscript which was carefully checked to avoid misunderstandings and conceptual errors. A certificate is attached.

Reviewer 2 Report

Paper well structured, figures and tables impressive. Observed overexpression of DGK in AML patients can really suggests the opportunity to further explore the biological role of DGK isoforms in AML and potential utility as therapeutic targets. Good work. 

Author Response

We thank the reviewer for his very positive comments. We have revised the paper according to the indications of other reviewers and undergone professional language editing.

Reviewer 3 Report

Acute myeloid leukemia (AML) is an aggresive tumor. Initial success concerning AML therapy based on FLT3, IDH1, IDH2, 37 and BCL2 inhibitors often disapears due to frequent occurrence of drug resistance. Diacylglycerol kinases (DGK) are a family of lipid signaling regulators found to be dissregulated in AML. The authors of this manuscript explored the DGKs family involvement in AML through a detailed analysis of expression databases and examinationof the in vitro effect of DGKs inhibitors on leukemia cell lines. They observed that selective DGKA inhibition (by CU-3 and AMB639752 compounds) was ineffective in killing AML cell lines, while wider specificity DGK inhibitors (R59022 and R59949) are effective in reducing the cell viability and arresting the cell cycle.

In general, the study is interesting. Unfortunately, the results of proposed selective inhibitors are not beneficial as hypothesized. I have several suggestion concerning the manuscript.

1. If the authors mean gene name it should be written in italics.

2. The last methodology section should have number 2.10, and not 3.0

3. In line 349 "+" is unnecessary.

4. Part D of Figure 3 is lacking description.

5. Figure 4: "Experiments were run in quadruplicates. Data are the mean ±SEM of 9 or more experiments interpolated as [inhibitor] vs. % viability." It is not clear how many experiments/technical replicates were made.

6. Please use full name "Figure" instead of Fig. in the main text.

Author Response

Acute myeloid leukemia (AML) is an aggresive tumor. Initial success concerning AML therapy based on FLT3, IDH1, IDH2, 37 and BCL2 inhibitors often disapears due to frequent occurrence of drug resistance. Diacylglycerol kinases (DGK) are a family of lipid signaling regulators found to be dissregulated in AML. The authors of this manuscript explored the DGKs family involvement in AML through a detailed analysis of expression databases and examination of the in vitro effect of DGKs inhibitors on leukemia cell lines. They observed that selective DGKA inhibition (by CU-3 and AMB639752 compounds) was ineffective in killing AML cell lines, while wider specificity DGK inhibitors (R59022 and R59949) are effective in reducing the cell viability and arresting the cell cycle.

In general, the study is interesting. Unfortunately, the results of proposed selective inhibitors are not beneficial as hypothesized. I have several suggestion concerning the manuscript.

  1. If the authors mean gene name it should be written in italics.

Done accordingly.

  1. The last methodology section should have number 2.10, and not 3.0

Done accordingly.

  1. In line 349 "+" is unnecessary.

Text revised.

  1. Part D of Figure 3 is lacking description.

Indeed, a formatting error occurred, and part of the legend was missing. Everting is corrected in this revised version.

  1. Figure 4: "Experiments were run in quadruplicates. Data are the mean ±SEM of 9 or more experiments interpolated as [inhibitor] vs. % viability." It is not clear how many experiments/technical replicates were made.

New legend: “Figure 4. Effects of DGK inhibitors on AML cell lines viability. A total of 50,000 cells were treated with increasing concentration of indicated inhibitor or vehicle for 24 h followed by alamarBlue viability assay for an additional 24 h. Data are presented as mean ± SEM of at least 9 independent experiments run in triplicates or quadruplicates, plotted as [inhibitor] vs. % viability.”.

  1. Please use full name "Figure" instead of Fig. in the main text.

Figure and Supplementary Figure written in the extended version.

Round 2

Reviewer 1 Report

The authors have addressed reviewers’ comments. In particular, the English language editing has been thorough and this has greatly assisted review of the manuscript. There are several minor errors that I have indicated in a copy of the manuscript and in addition, I have included suggested edits to the legends for Figures 5 and 6 to remove methods descriptions, which are already detailed in the Methods section and do not need to be repeated. I still feel that some interpretations of results are overstated. The authors have provided no evidence that inhibition of DGKA would be inhibitory to AML cells (in fact, their cell line experiments indicates that DGKA inhibition would fail to inhibit AML cells). However, they still conclude that DGKA would be an ‘attractive therapeutic target’ (Line 532). Similarly, the lack of correlation of results between databases (TCGA vs BeatAML) provides no support for an association between any DGK isoform and AML outcomes. As such, phrases such as ‘consistent with their divergent effects on patients’ survival’ (Line 387) are not warranted. Even where gene expression can be reproducibly correlated with outcomes, additional information will always be required to indicate whether the gene of interest is a driver or bystander in this process. Despite my strong reservations regarding the authors’ interpretations of their data, I feel that the data are presented in a comprehensive manner and that readers would be able to independently determine whether they agree with the authors’ interpretations. For these reasons, I feel that the manuscript is suitable for publication, pending correction of minor errors indicated below and in the attached copy of the manuscript.

1. Please indicate whether the bars are s.d. or s.e.m. in the legend for Figure 5.

2. Gene names are usually written in italics, while the encoded protein is written in normal lettering.

Author Response

The authors have addressed reviewers’ comments. In particular, the English language editing has been thorough and this has greatly assisted review of the manuscript.

Thanks a lot for the appreciation.

There are several minor errors that I have indicated in a copy of the manuscript

We are really grateful for the careful revision and corrections that ameliorated the manuscript and were all accepted in this revised version

and in addition, I have included suggested edits to the legends for Figures 5 and 6 to remove methods descriptions, which are already detailed in the Methods section and do not need to be repeated.

We have revised the indicated legends to remove methodology while maintaining clarity for the reader.

I still feel that some interpretations of results are overstated. The authors have provided no evidence that inhibition of DGKA would be inhibitory to AML cells (in fact, their cell line experiments indicates that DGKA inhibition would fail to inhibit AML cells). However, they still conclude that DGKA would be an ‘attractive therapeutic target’ (Line 532).

Revised to “Collectively, these data suggest that DGKs are involved in the differentiation and cell cycle progression of leukaemia cells in an isoform-specific fashion and point to DGKs as attractive therapeutic targets.”

Similarly, the lack of correlation of results between databases (TCGA vs BeatAML) provides no support for an association between any DGK isoform and AML outcomes. As such, phrases such as ‘consistent with their divergent effects on patients’ survival’ (Line 387) are not warranted. Even where gene expression can be reproducibly correlated with outcomes, additional information will always be required to indicate whether the gene of interest is a driver or bystander in this process. Despite my strong reservations regarding the authors’ interpretations of their data, I feel that the data are presented in a comprehensive manner and that readers would be able to independently determine whether they agree with the authors’ interpretations.

Indeed, we are conscious that this is our interpretation of the data and appreciate that the reviewer, although not agreeing with it, feels that is correctly presented. We aim to stimulate further research in the field.

For these reasons, I feel that the manuscript is suitable for publication, pending correction of minor errors indicated below and in the attached copy of the manuscript.

Thanks, we have done accordingly.

  1. Please indicate whether the bars are s.d. or s.e.m. in the legend for Figure 5.

SD as now indicated in figure 5 and in supplementary materials 8.

  1. Gene names are usually written in italics, while the encoded protein is written in normal lettering.

As suggested, we carefully revised the text to write gene names and RNA in italics and protein in normal lettering.